# PLDP-FL: Federated Learning with Personalized Local Differential Privacy

**DOI:** 10.3390/e25030485

**Published:** 2023-03-10

**Authors:** Xiaoying Shen, Hang Jiang, Yange Chen, Baocang Wang, Le Gao

**Affiliations:** 1The State Key Laboratory of Integrated Service Networks, Xidian University, Xi’an 710071, China; 2The Key Laboratory of Cryptography of Zhejiang Province, Hangzhou Normal University, Hangzhou 310030, China; 3The School of Information Engineering, Xuchang University, Xuchang 461002, China; 4The Faculty of Intelligent Manufacturing, Wuyi University, Jiangmen 529000, China

**Keywords:** federated learning, privacy protection, local differential privacy, personalization

## Abstract

As a popular machine learning method, federated learning (FL) can effectively solve the issues of data silos and data privacy. However, traditional federated learning schemes cannot provide sufficient privacy protection. Furthermore, most secure federated learning schemes based on local differential privacy (LDP) ignore an important issue: they do not consider each client’s differentiated privacy requirements. This paper introduces a perturbation algorithm (PDPM) that satisfies personalized local differential privacy (PLDP), resolving the issue of inadequate or excessive privacy protection for some participants due to the same privacy budget set for all clients. The algorithm enables clients to adjust the privacy parameters according to the sensitivity of their data, thus allowing the scheme to provide personalized privacy protection. To ensure the privacy of the scheme, we have conducted a strict privacy proof and simulated the scheme on both synthetic and real data sets. Experiments have demonstrated that our scheme is successful in producing high-quality models and fulfilling the demands of personalized privacy protection.

## 1. Introduction

In recent years, artificial intelligence (AI) technology has made great strides, bringing about positive changes in areas such as finance, healthcare, and education [1]. However, owing to the heightened awareness of privacy protection and the implementation of laws and regulations, the problem of data islands has become increasingly noticeable, thus impeding the training of AI models. The introduction of Federated Learning (FL) [2,3] has been a critical technology for AI development, as it allows sharing data between multiple parties while guaranteeing user privacy and data security. Federated learning aims to establish a multi-party collaborative machine learning (ML) model, where the data of participants are kept in their local area, and only the intermediate parameters are exchanged and transferred to train the final model.

It has been noted in prior studies [4,5] that federated learning cannot provide sufficient privacy protection. External adversaries and internal adversaries can still infer clients’ private information through gradient parameters or model parameters, such as model inversion attacks [6] and membership inference attacks [7]. Currently, there are many approaches to achieve secure federated learning, such as a combination of federated learning and secure multi-party computation (SMC) [8] or homomorphic encryption (HE) [9] schemes. They can offer robust protection measures. However, they also incur heavy computational and communication overhead. Differential privacy (DP) [10], a lightweight privacy protection technology with low computational and communication costs, has been widely applied in machine learning to protect privacy. A federated optimization algorithm for client-level differential privacy preservation has been presented in [11], which aims to hide the client’s contribution during the training process, while still achieving a balance between privacy loss and model performance. Local differential privacy has been suggested in [12,13,14] to address the privacy leakage problem in federated learning. As federated learning and local differential privacy are suitable for distributed architectures, their integration is a natural choice. Meanwhile, local differential privacy can ensure stringent privacy protection for federated learning, and effectively resist membership inference attacks.

However, the above-mentioned federated learning schemes based on local differential privacy have a common problem, which is that they assume that all local clients have the same privacy protection requirements, thus each local client has the same privacy budget. Considering the differences in privacy preferences and data sensitivity of each client in the real world, simply setting the same level of privacy protection will lead to a situation where some clients have insufficient privacy protection and some clients have excessive privacy protection. In light of this, this paper proposes a federated learning scheme based on personalized local differential privacy, to satisfy the local clients’ differential privacy requirements. In summary, our main contributions are as follows:**Personalized perturbation mechanism**. We provide a novel personalized local differential privacy data perturbation mechanism. By adjusting the privacy budget parameters and introducing distinct security range parameters for each client, this perturbation mechanism enables clients to articulate their different privacy requirements and obtain various privacy protection.**Privacy-preserving framework**. We propose a privacy-preserving federated learning method to address the demand for personalized local differential privacy. By employing the personalized differential privacy perturbation algorithm to process the intermediate parameters, the server cannot deduce the participant’s privacy information from the intermediate parameters, thereby achieving the personalized privacy protection requirements in the federated learning process.**High-quality model**. Extensive experimental results show that our scheme can obtain a high-quality model when the clients have privacy-preserving requirements. Thus, it is more suitable for applications with device heterogeneity in the real world.

**Organization.** In the following, we introduce related work in Section 2. Then, we introduce the preliminaries in Section 3. Section 4 gives the system overview and Section 5 gives the details of the PLDP-FL. The performance evaluation and experimental results are shown in Section 6. Section 7 concludes the paper. A summary of basic concepts and notations is provided in Table 1.

## 2. Related Work

Federated learning ensures the security of the original data by transferring only the intermediate calculation results. However, in some cases, the original data can still be retrieved if the intermediate parameters are attacked [4,5,6,7]. There are still security risks associated. To bridge the gap of common federated learning techniques, secure federated learning has been proposed in academia and industry. Cryptographic schemes based on homomorphic encryption and secure multi-party computation [8,9,15,16,17] can safeguard the intermediate parameters and results from attacks. The schemes based on differential privacy [11,12,13,14,18,19,20,21,22,23,24,25,26,27] ensure the security of process and result, but also introduce a certain amount of computational error.

**Cryptographic techniques**. Truex et al. [8] combined secure multi-party computation and differential privacy to defend against membership inference attacks and protected data privacy in federated learning. Threshold pailier encryption is employed to implement secure multi-party computation [28]. The noise of differential privacy injection is reduced by introducing a trusted parameter, while solving the problem of the existence of untrustworthy clients. Xu et al. [16] to further reduce the communication costs in [8], they combined a secure multi-party computation protocol implemented based on function encryption and differential privacy to protect data privacy in federated learning. Xu et al. [9] proposed a federated deep learning framework to address the issue of untrusted clients, utilizing Yao’s obfuscation circuit and additive homomorphic encryption to protect the private data of clients. Phong et al. [15] combined additive homomorphic encryption and stochastic gradient descent algorithm to ensure that the server will not be aware of the client’s private information. The homomorphic encryption method in their scheme provided two implementations, one based on LWE and the other based on Pallier.

**Differential privacy**. Cryptography-based schemes suffer from low computational performance and high communication costs, so methods based on differential privacy technology have received increasing attention. Shokri et al. [18] first applied differential privacy to privacy protection in deep learning. Nevertheless, it required a large amount of privacy budget to ensure the model’s utility. Thus its privacy protection is not very adequate. Subsequently, Abadi et al. [19] also employed differential privacy to protect the gradient information in the random gradient descent algorithm, and proposed a novel algorithm to improve the privacy cost in the learning process. Hu et al. [20] considered the practical problem of user heterogeneity in the distributed learning system, and proposed a privacy-preserving method that satisfies differential privacy, which is used to learn a personalized model on distributed user data.

**Local differential privacy**. Chamikara et al. [27] proposed a new local differential privacy algorithm using a random response technique, which was inserted between the convolution layer and full connection layer of the convolutional neural network during training, to achieve privacy protection by perturbing the binary encoding of the input data. Truex et al. [12] proposed a new federated learning system for protecting privacy in deep neural networks by using a variant of the LDP algorithm CLDP. Although their approach can handle complex models, its corresponding privacy budget parameter is too large to be practical. Sun et al. [26] introduced a noise-free differential privacy concept that employs sampling (with or without replacement) of the client’s data samples during the FL process, achieving improved performance. Sun et al. [14] proposed a federated learning scheme based on local differential privacy. This approach considers different value ranges of model parameters, and designs corresponding perturbation algorithms to ensure parameter privacy. At the same time, this scheme introduces a parameter transformation process that rearranges and shuffles the parameters uploaded by the client, thus providing enhanced privacy protection. However, the shuffling mechanism in this method is equivalent to introducing another third party, which not only increases the security burden of the system, but also increases the communication overhead of the system.

**Personalized privacy models**. With the emerging demand for privacy, personalized privacy research [29,30,31,32,33,34] has attracted more and more attention. Nie et al. [29] first proposed a utility optimization framework, for histogram estimation with personalized multilevel privacy. To clarify the goal of privacy protection, they personalized the traditional definition of LDP. Shen et al. [30] considered the personality of data owners in protecting and utilizing multidimensional data by introducing the concept of personalized local differential privacy. They designed personalized multiple optimized unary encoding to perturb data owners’ data, and proposed an aggregation algorithm for frequency estimation of multidimensional data under personalized local differential privacy. Xue et al. [31] used a personalized local differential privacy model based on previous mean estimation schemes under LDP to design novel methods to provide users with personalized privacy. Yang et al. [34] first proposed an algorithm PLU-FedOA to optimize horizontal federated learning with personalized local differential privacy, which allows clients to select the privacy level. Nevertheless, this method has two drawbacks. Firstly, the method only has one privacy parameter, and the range of parameters and parameter values is too wide, resulting in inadequate privacy protection. Secondly, the method enforces differential privacy protection by incorporating Laplace noise. When the privacy budget parameter is small, the Laplace noise will generate a large variance, thus significantly impacting the model performance. To solve the above problems, we propose a novel federated learning scheme based on personalized local differential privacy.

## 3. Preliminaries

### 3.1. Local Differential Privacy

Differential privacy [35] is a privacy-preserving technique that does not limit adversary capabilities, and can accurately define and evaluate privacy protection quantitatively. Differential privacy can ensure two security objectives: first, the adversary cannot carry out the link attack. Second, even if the attacker can verify that the target is in the data set, they will not be able to gain accurate information about the target through the differential output result. Traditional differential privacy, also known as global differential privacy (GDP), and its implementation depends on a trusted third party. However, in reality, it is difficult to find a completely trusted third party, so there are limitations in applying global differential privacy. Local differential privacy is a distributed variant of GDP that allows each client to perturb its data locally and then upload them to the server, so the server never has access to the client’s real data.

**Definition 1** ((ϵ-LDP) [36]). *Assuming that M is a randomized algorithm, D and D′ are two separate datasets with only one dissimilar datum, if and only if for any output O∈ Range(M) of the randomized algorithm M, the following inequality is hold.*
(1)$$\mathrm{Pr}\left[M\left(v\right)=O\right]\leq e^{\epsilon }\cdot \mathrm{Pr}\left[M\left(v^{\prime }\right)=O\right],$$
*The randomized algorithm M is said to satisfy ϵ-LDP, where ϵ≥0, Range(M) denotes the set of all possible outputs of M, Pr[·] denotes the probability, and ϵ denotes the privacy budget.*

Differential privacy has become one of the more popular privacy-preserving techniques due to its remarkable features, such as post-processing invariance and composability, which are essential for its transition from theory to practical application.

**Proposition 1** (Post-processing).
*Given a randomized algorithm $M_{1}$ satisfies ϵ-DP, for any randomized algorithm $M_{2}$, the combination $M_{2}\left(M_{1}\right)$ of $M_{1}$ and $M_{2}$ satisfies ϵ-DP.*


**Proposition 2** (Sequential composition).
*Assuming that each $M_{i}$ provides $\epsilon_{i}$-DP, the sequence combination of $M_{i}\left(D\right)$ provides $\sum_{i}\epsilon_{i}$-DP. Where $M_{i}$ denotes a differential privacy algorithm satisfies $\epsilon_{i}$, and $M_{i}\left(D\right)$ denotes all the algorithms act on the same dataset D.*


### 3.2. Personalized Local Differential Privacy

A significant characteristic of local differential privacy is that individuals can independently disturb data to protect their privacy. As participants are the ones who understand their privacy requirements best, there may have varying privacy requirements dependent on the sensitivity of their data. Therefore, it is necessary to promote the concept of personalized privacy in local settings. Chen et al. [37] suggested a novel personalized local differential privacy model for learning the distribution of users’ spatial data. This paper utilizes the same method found in [37] to characterize the various privacy preferences of clients.

**Definition 2** (((τ,ϵ)-PLDP) [37]). *Assuming that M is a randomized algorithm, given the personalized privacy parameters (τ,ϵ) of the client and any pair of input values $v,v^{\prime }\in \tau$, if and only if for any output O⊆ Range(M) of the randomized algorithm M, the following inequality holds.*
(2)$$\mathrm{Pr}\left[M\left(v\right)\in O\right]\leq e^{\epsilon }\cdot \mathrm{Pr}\left[M\left(v^{\prime }\right)\in O\right].$$
*Then the randomized algorithm M is said to satisfy (τ,ϵ)-PLDP, where τ denotes safe range, Range(M) denotes the set of all possible outputs of M, Pr[·] denotes the probability, and ϵ denotes the privacy budget.*

### 3.3. Random Response

#### 3.3.1. Random Response

Random response (RR) [38] began as a method for sampling and has since become a popular technology for implementing local differential privacy. The main idea of random response is to protect users’ privacy by providing a plausible answer to the sensitive question. Flipping a coin is the most common method for answering a binary value. Utilizing coin flip technology, it can answer privacy-related questions. If the result is positive, the user will answer truthfully, and a second coin flip will be executed if the result is negative. If the result is positive, the answer is yes, and if the result is negative, the answer is no. For the data collector, he knows the probability of the user’s true answer. For the user, the data collector does not know the value of his answer, so the user’s privacy is protected to some extent.

This random response mechanism fulfills (ln3,0) differential privacy. It is obvious that the probability of users giving a YES answer when the true value is Yes is 0.75, and the chance of a YES response when the real value is No is 0.25. When the response result is determined, the following equation holds.
(3)$$\frac{\mathrm{Pr}[Ans=YES|Truth=Yes]}{\mathrm{Pr}[Ans=YES|Truth=No]}=3.$$

We can also understand the random response in another way, i.e., the user having a probability of *p* to answer the true value and a probability of 1−p to answer the opposite value. Given a response result, we can obtain the ratio between the probability of responding to the true result and the probability of responding to other results $e^{\epsilon }=\frac{p}{1-p}$. Further, we get $p=\frac{e^{\epsilon }}{e^{\epsilon }+1}$, it can provide $ln(\frac{p}{1-p})$-LDP. To ensure that the privacy budget is greater than 0, it is generally recommended that the value of *p* be higher than 0.5.

#### 3.3.2. Generalized Random Response

Generalized random response(GRR) [39] is a generalization of the RR for the case of answering multiple choice, also known *k*-RR. The idea is similar to that of RR, i.e., there is a large probability of providing an honest answer, whereas the probability of giving a false answer is small. On the one hand, The total of all probability values should be 1. On the other hand, the ratio of large probability to small probability needs to satisfy the definition of ϵ−LDP, where $\epsilon =ln(\frac{(k-1)p}{1-p})$. Currently, the values of *p* and *q* fulfill the following equation under optimal local difference privacy.
(4)$$\mathrm{Pr}\left[M\left(v\right)=\tilde{v}\right]=\left\{\begin{array}{c} p=\frac{e^{\epsilon }}{e^{\epsilon }+k-1},\;\mathrm{if}\;\tilde{v}=v,\;\\ q=\frac{1}{e^{\epsilon }+k-1},\;\mathrm{if}\;\tilde{v}\neq v, \end{array}\right.$$
where *v* is the original input value, $\tilde{v}$ is the perturbed value, *p* is the probability of obtaining the true response, and *q* is the probability of responsing the other values.

### 3.4. Federated Deep Learning

Federated learning is a distributed machine learning approach that allows sharing of data and training models while protecting privacy. Google’s McMahan et al. [2] pioneered the use of federated learning to modify the language prediction model on smartphones. This approach focuses on achieving a balance between data privacy protection and sharing by exchanging the intermediate parameters instead of the original data when multiple participants collaborate to train the model.

Federated Average Algorithm (FedAvg) is the core algorithm employed in federated learning. FL generally involves local clients and parameter server, and each client possesses a private dataset $D_{i}={\left(x_{i},y_{i}\right)}_{i=1}^{N}$, where $x_{i}$ and $y_{i}$ are the input data and true label, respectively, and *N* is the number of clients. Suppose the loss function of a learning task is L(w;x,y), where *w* is the weight parameter, *x* and *y* are the training sample. The learning goal is to construct an empirical minimization such as $w^{*}=argmin_{w}\frac{1}{N}\sum_{i=1}^{N}L\left(w;x_{i},y_{i}\right)$ on data from *N* clients. During each round of t∈[0,T], the participant $P_{i}$ performs the model update in parallel on the local dataset using the stochastic gradient descent (SGD) algorithm,
(5)$$w_{t+1}^{i}=w_{t}^{i}-\eta \frac{\partial L_{i}}{\partial w_{t}^{i}}.$$

Then sends the local update parameters to the server for global aggregation,
(6)$${\bar{w}}_{t+1}=\frac{1}{N}\sum_{i=1}^{N}w_{t+1}^{i}.$$

Finally, after aggregating the model parameters, send them back to the client and repeat the process until either the maximum number of training rounds *T* is reached or the model converges.

## 4. System Overview

### 4.1. Problem Definition

This paper focuses on the federated learning problem in the context of personalized local differential privacy. Its goal is to collaborate with *N* clients and a semi-honest server to train a global model, while respecting the varying levels of privacy protection desired by each participant. Each client has pairs of privacy parameters ($\tau_{i},\epsilon_{i}$) that characterize its privacy preferences. $\epsilon_{i}$ is the privacy budget, and $\tau_{i}$ reflects the personalized privacy requirements. Without loss of generality, if the client’s original data range is assumed to be [−1, 1] and is uploaded to the server without any processing, it is vulnerable to certain attacks that could expose the victim’s private information. This paper ensures the security of the entire process by introducing personalized local differential privacy to perturb the data uploaded by the client.

### 4.2. System Model

Federated learning scheme based on personalized local differential privacy (PLDP-FL) involves local clients and parameter server, and its system model is shown in Figure 1. In the following, we will explain the functions of each entity within the system.

**Local clients:** Assume that there are *N* local clients $P=\{P_{i}|1\leq i\leq N\}$ in the system, and each client $P_{i}$ possesses its private dataset, while they have full authority over their local data. During each round of iterative updates, clients use their local data for model parameter updates, and then upload them to the server for global updates. To ensure the security of their private data during the interaction of intermediary parameters, each client $P_{i}$ perturbs the intermediate parameters using the local differential privacy algorithm before uploading them to the server. At the same time, each client $P_{i}$ can set its own desired privacy parameters according to the sensitivity of its data and privacy preference. Finally, each client receives the update results from the server for a new round of local updates.

**Parameter server:** Assuming that the system is only equipped with a parameter server, it is endowed with powerful computing and storage capabilities. This server acts as a collaborator among all participants, receiving the intermediate parameters uploaded by them for global update operations and enabling data sharing among them.

### 4.3. Threat Model

In this scenario, we assume that the parameter server is honest and curious, meaning it will strictly execute the process of federated learning accurately, and return the correct results to all clients after global aggregation of the received intermediate parameters. Furthermore, it will be curious to the participants’ private information, and it can obtain this information by analyzing the participants’ intermediate parameters. All local participants are expected to be honest and curious. They will adhere to the model training process and not disrupt the learning process with malicious intent. Additionally, they shall attempt to acquire the private information of other local participants through aggregation results or public channels. We further assume that the parameter server will collaborate with some local participants, but only with some. There is an active adversary A in our proposed scheme. The goal of A is to infer the private information of participants by utilizing intermediate parameters or aggregation results. Specifically, the adversary A in PLDP-FL has the following capabilities.

The adversary A can eavesdrop on intermediate parameters in the interaction of each entity via the public channel, thus inferring the client’s private information from them.The adversary A can compromise the parameter server, and exploit the server’s intermediate parameters or aggregation results to deduce the private information of the local participant.The adversary A can corrupt one or more local participants, and use their information to infer the privacy details of other local participants. It is not permissible to corrupt all local participants simultaneously.

### 4.4. Design Goal

According to the above system model and threat model. The proposed scheme must ensure data privacy, local participants’ personalized privacy functionality, and the model’s accuracy.

**Data privacy**: Our scheme should guarantee that adversaries cannot access local participants’ private data, either directly or by using the intermediate parameters and aggregated results.**Local participants’ personalized privacy functionality** Considering the varying sensitivities of each participant’s private data and their privacy preferences. Our scheme should ensure that all participants can adjust their privacy parameters to satisfy their personalized privacy protection requirements.**Model’s accuracy** Our scheme should ensure that the model converges in theory, while also ensuring its practical feasibility, i.e., the privacy-protected and non-privacy-protected federated learning should be able to train models that are almost the same.

## 5. Proposed Scheme

In this section, we will discuss the implementation details of PLDP-FL, including the steps of implementation and the critical personalized perturbation algorithm. Specifically, the tasks of the parameter server and local participants, as well as the implementation steps of the scheme, will be outlined, and a corresponding implementation flow chart will be presented. Moreover, the perturbation algorithm is discussed in detail, including its algorithm description and privacy proof.

### 5.1. Steps of Implementation

The PLDP-FL scheme involves *N* local participants $P=\{P_{1},P_{2},\ldots ,P_{N}\}$, and a parameter server. Each participant has their own private data set, with varying privacy sensitivity. The aim is to collaboratively train a global model with their local data, without compromising any privacy. Algorithm 1 describes the overall process of PLDP-FL, which mainly consists of two phases: server update and client update.
**Algorithm 1** PLDP-FL**Require:** *N* is the number of local clients, γ is the local client selection factor, 0<γ≤1, *E* is the number of local epochs, *B* is the local mini-batch size, η is the learning rate, L is the loss function.
**Ensure:** The trained model *W*.
 1:**ServerUpdate:** 2:Initialize the weight parameter $w_{0}$ and send it to all clients; 3:**for** each round t=1,2,…,T
**do** 4:   The server randomly selects γ·N local clients $P_{t}$; 5:   **for** each client $P_{i\in [N]}$ in parallel **do** 6:     $w_{i,t}\leftarrow$ ClientUpdate $\left(P_{i},{\bar{w}}_{t-1}\right)$ and sends the result to the server; 7:   **end for** 8:   ${\bar{w}}_{t}\leftarrow \frac{1}{|P_{t}|}\sum_{i}w_{i,t};$ 9:**   return** ${\bar{w}}_{t}$. 10:**end for** 11:**ClientUpdate$\left(P_{i},{\bar{w}}_{t-1}\right)$:** 12:t←t+1; 13:Receive the latest updates from the server: $w_{i,t}\leftarrow {\bar{w}}_{t-1}$; 14:**for** each local epoch j=1,2,…,E
**do** 15:   **for** each batch b∈B
**do** 16:      $w_{i,t}\leftarrow w_{i,t}-\eta \cdot \nabla L\left(w_{i,t};b\right)$; 17:   **end for** 18:**end for** 19:${\tilde{w}}_{i,t}\leftarrow$Personalized($w_{i,t}$).


#### 5.1.1. Server Update

First, the server initializes the global model parameter $w_{0}$ and distributes it to all clients, thus providing them with the same model structure. During each round of interaction, the server randomly selects γ·N local clients to upload model parameter, instead of selecting all clients. The large number of local clients results in a significant communication overhead when sending parameters to the server, and this overhead increases with the number of clients. Therefore, randomly selecting a certain proportion of clients for parameter sharing can decrease the communication overhead. For the aggregation operation, we use the gradient average method
(7)$${\bar{w}}_{t}\leftarrow \frac{1}{|P_{t}|}\sum_{i}w_{i,t}\;,$$
where $|P_{t}|$ indicates the number of randomly chosen participants and $w_{i,t}$ denotes the model parameters of participants $P_{i}$ in round *t*. Owing to the local personalized differential privacy perturbation algorithm being unbiased, the server does not need to carry out any further operations after the aggregation operation. After the aggregation is completed, the server will check if the model has converged or if the maximum number of communication rounds has been reached. If either of these conditions is fulfilled, the server will notify the participants to stop training and send the final model to all participants. Otherwise, it will transmit the aggregation results to the participants to continue training.

#### 5.1.2. Client Update

Upon receiving the latest model parameters from the server, each participant replaces them with the model parameters of the current round. Then, they perform forward and back propagation to update the parameters. A total of *E* iterations are required to be executed locally. During each iteration, *B* training samples are randomly chosen. In forward propagation, the training data are extracted and compressed sequentially via the stacked convolution, ReLU, and pooling layers. The training sample’s prediction label value is finally obtained through the full connection layer. Then, calculate the cross entropy loss function value $L_{i,t}$ by the predicted label values and their corresponding true label values,
(8)$$L_{i,t}=\frac{1}{B}\sum_{i}-\left[y_{i}ln\left(p_{i}\right)+\left(1-y_{i}\right)ln\left(1-p_{i}\right)\right].$$
where $y_{i}$ represents the true label value of the sample, while $p_{i}$ denotes the predicted label value of the sample. In backward propagation, the gradient value of the weight parameter is obtained by calculating the partial derivative of the weight parameter in relation to the loss value. Then the weight parameter is updated through the small batch gradient descent algorithm,
(9)$$\omega_{i,t}=\omega_{i,t}-\eta \cdot ▽L\left(\omega_{i,t};b\right),$$
where $▽L(\omega_{i,t};b)$ denotes the gradient value.

Since the server is not completely honest, the updated weight parameters of each client cannot be uploaded directly. Therefore, we employ the personalized local differential privacy algorithm to perturb them before uploading. Hereafter, we will discuss the detailed process of the perturbation algorithm.

### 5.2. Personalized Perturbation Algorithm

The existing local differential privacy protection mechanism has a significant limitation that assumes all clients have the same privacy protection level. That is, the same privacy budget is set. Since each client has different data sensitivity requirements, data privacy protection can either be inadequate or excessive. Inspired by the ideas of Chen et al. [37] and Xue et al. [31], we propose a novel perturbation algorithm PDPM that satisfies personalized local differential privacy. Its pseudocode is shown in Algorithm 2. The PDPM algorithm involves two stages.
**Algorithm 2** PDPM**Require:** The personalized privacy parameters $\left(\tau_{i},\epsilon_{i}\right)$, the range length of $\tau_{i}$ is $L_{i}=\left|\tau_{i}\right|$, the midpoint of $\tau_{i}$ is $c_{i}$, and the client’s input value $w_{i}\in \tau_{i}$.
**Ensure:** The value after perturbation ${\tilde{w}}_{i}$.
 1:**if** 
$c_{i}\neq 0$ 
**then** 2:   Translating $c_{i}$ to point 0 so that $\tau_{i}$ is symmetric about point 0; 3:   $v_{i}=w_{i}-c_{i}$; 4:**else** 5:   $v_{i}=w_{i}$. 6:**end if** 7:Sampling a random variable *a* such that 8:      $Pr\left[a=L_{i}/2\right]=\frac{v_{i}\left(e^{\epsilon_{i}}-1\right)}{L_{i}\left(e^{\epsilon_{i}}+2\right)}+\frac{e^{\epsilon_{i}}+1}{2(e^{\epsilon_{i}}+2)}$; 9:**if** 
$a=L_{i}/2$ 
**then** 10:   ${\tilde{v}}_{i}=\frac{L_{i}\left(e^{\epsilon_{i}}+3\right)}{2(e^{\epsilon_{i}}-1)}$; 11:**else if** 
$a=-L_{i}/2$
**then** 12:   ${\tilde{v}}_{i}=-\frac{L_{i}\left(e^{\epsilon_{i}}+1\right)}{e^{\epsilon_{i}}-1}$; 13:**else** 14:   ${\tilde{v}}_{i}=0$; 15:**end if** 16:${\tilde{w}}_{i}={\tilde{v}}_{i}+c_{i}$; 17:**return** ${\tilde{w}}_{i}$.


#### 5.2.1. Set Privacy Parameter

Each participant sets their privacy parameter pair $\left(\tau_{i},\epsilon_{i}\right)$, where $\tau_{i}$ denotes the security interval, the client’s true value $w_{i}\in \tau_{i}$, $\tau_{i}$ is the minimum interval in which the clients’ expected true value is indistinguishable, and $\epsilon_{i}$ is privacy budget that indicates the privacy protection level. Let $L_{i}$ denote the length of $\tau_{i}$ and $c_{i}$ represents the center of $\tau_{i}$. As each participant can define their privacy parameters, their input value range varies, increasing the complexity of implementing the perturbation algorithm. To achieve a unified perturbation algorithm, we request that each participant first determine if the center $c_{i}$ of the security interval is at zero. If so, no operation is necessary. Otherwise, the center $c_{i}$ of the security interval will be shifted to zero, producing a symmetrical security interval. The original data are correspondingly translated to $v_{i}=w_{i}-c_{i}$, so that each client’s input value $v_{i}\in \left[-L_{i}/2,L_{i}/2\right]$.

#### 5.2.2. Compute Perturbation Value

The perturbation value ${\tilde{v}}_{i}$ of $v_{i}$ is obtained for each client according to the following probabilities.
(10)$$\mathrm{Pr}\left[M\left(v_{i}\right)={\tilde{v}}_{i}\right]=\left\{\begin{array}{c} \frac{v_{i}\left(e^{\epsilon_{i}}-1\right)}{L_{i}\left(e^{\epsilon_{i}}+2\right)}+\frac{\left(e^{\epsilon_{i}}+1\right)}{2\left(e^{\epsilon_{i}}+2\right)},\;\mathrm{if}\;{\tilde{v}}_{i}=\frac{L_{i}\left(e^{\epsilon_{i}}+3\right)}{2\left(e^{\epsilon_{i}}-1\right)},\;\\ \frac{e^{\epsilon_{i}}+3}{4\left(e^{\epsilon_{i}}+2\right)}-\frac{v_{i}\left(e^{\epsilon_{i}}-1\right)}{2L_{i}\left(e^{\epsilon_{i}}+2\right)},\;\mathrm{if}\;{\tilde{v}}_{i}=\frac{-L_{i}\left(e^{\epsilon_{i}}+1\right)}{e^{\epsilon_{i}}-1},\;\\ \frac{e^{\epsilon_{i}}+3}{4\left(e^{\epsilon_{i}}+2\right)}-\frac{v_{i}\left(e^{\epsilon_{i}}-1\right)}{2L_{i}\left(e^{\epsilon_{i}}+2\right)},\;\mathrm{if}\;{\tilde{v}}_{i}=0, \end{array}\right.$$
where M denotes the perturbation algorithm. The algorithm produces the corresponding disturbance value under this probability. Since the perturbation is applied to the translated value ${\tilde{v}}_{i}$, the original value ${\tilde{w}}_{i}={\tilde{v}}_{i}+c_{i}$ is obtained.

**Theorem 1.** 
*Algorithm 2 provides $\left(\tau_{i},\epsilon_{i}\right)$-PLDP for each client. In addition, the perturbation of algorithm satisfy unbiasedness, i.e., $E\left({\tilde{w}}_{i}\right)=w_{i}$, and $Var\left[{\tilde{w}}_{i}\right]\leq \frac{\left(e^{\epsilon_{i}}+1\right)\left(e^{\epsilon_{i}}+3\right)}{L_{i}{\left(e^{\epsilon_{i}}-1\right)}^{2}}$.*


**Proof.** For any ${\tilde{w}}_{i}\in \left\{\frac{L_{i}\left(e^{\epsilon_{i}}+3\right)}{2(e^{\epsilon_{i}}-1)}+c_{i},-\frac{L_{i}\left(e^{\epsilon_{i}}+1\right)}{e^{\epsilon_{i}}-1}+c_{i},c_{i}\right\}$, and any two values $w_{i},w_{i}^{\prime }\in \tau_{i}$, we have $v_{i}=w_{i}-c_{i}$, $v_{i}^{\prime }=w_{i}^{\prime }-c_{i}$, ${\tilde{v}}_{i}={\tilde{w}}_{i}-c_{i}$. Then,
(11)$$\begin{array}{cc} \frac{\mathrm{Pr}\left[{\tilde{w}}_{i}|w_{i}\right]}{\mathrm{Pr}\left[{\tilde{w}}_{i}|w_{i}^{\prime }\right]} & =\frac{\mathrm{Pr}\left[{\tilde{v}}_{i}|v_{i}\right]}{\mathrm{Pr}\left[{\tilde{v}}_{i}|v_{i}^{\prime }\right]}\\ \; & =\frac{2v_{i}\left(e^{\epsilon_{i}}-1\right)+L_{i}\left(e^{\epsilon_{i}}+1\right)}{2v_{i}^{\prime }\left(e^{\epsilon_{i}}-1\right)+L_{i}\left(e^{\epsilon_{i}}+1\right)}\\ \; & \leq \frac{L_{i}\left(e^{\epsilon_{i}}-1\right)+L_{i}\left(e^{\epsilon_{i}}+1\right)}{-L_{i}\left(e^{\epsilon_{i}}-1\right)+L_{i}\left(e^{\epsilon_{i}}+1\right)}\\ \; & =e^{\epsilon_{i}}. \end{array}$$ Thus Algorithm 2 satisfies $\left(\tau_{i},\epsilon_{i}\right)$-PLDP. Next,
(12)$$\begin{array}{cc} E\left({\tilde{w}}_{i}\right) & =E\left({\tilde{v}}_{i}+c_{i}\right)\\ \; & =E\left({\tilde{v}}_{i}\right)+c_{i}\\ \; & =\frac{L_{i}\left(e^{\epsilon_{i}}+3\right)}{2\left(e^{\epsilon_{i}}-1\right)}\cdot \left[\frac{v_{i}\left(e^{\epsilon_{i}}-1\right)}{L_{i}\left(e^{\epsilon_{i}}+2\right)}+\frac{\left(e^{\epsilon_{i}}+1\right)}{2\left(e^{\epsilon_{i}}+2\right)}\right]-\\ \; & \frac{L_{i}\left(e^{\epsilon_{i}}+1\right)}{e^{\epsilon_{i}}-1}\cdot \left[\frac{e^{\epsilon_{i}}+3}{4\left(e^{\epsilon_{i}}+2\right)}-\frac{v_{i}\left(e^{\epsilon_{i}}-1\right)}{2L_{i}\left(e^{\epsilon_{i}}+2\right)}\right]+c_{i}\\ \; & =v_{i}+c_{i}\\ \; & =w_{i}. \end{array}$$ Moreover, for the mean value of the perturbed data,
(13)$$\begin{array}{cc} E(\bar{M(w)}) & =E\left(\frac{1}{\gamma N}\sum_{i=1}^{\gamma N}{\tilde{w}}_{i}\right)\\ \; & =\frac{1}{\gamma N}\sum_{i=1}^{\gamma N}E\left({\tilde{w}}_{i}\right)=\bar{w}. \end{array}$$
where $\bar{M(w)}=\tilde{w}$, and then,
(14)$$\begin{array}{cc} \mathrm{Var}\left[{\tilde{w}}_{i}\right] & =\mathrm{Var}\left[{\tilde{v}}_{i}+c_{i}\right]\\ \; & =\mathrm{Var}\left[{\tilde{v}}_{i}\right]\\ \; & =E\left({\tilde{v}}_{i}^{2}\right)-{\left(E\left({\tilde{v}}_{i}\right)\right)}^{2}\\ \; & \leq \frac{\left(e^{\epsilon_{i}}+1\right)\left(e^{\epsilon_{i}}+3\right)}{L_{i}{\left(e^{\epsilon_{i}}-1\right)}^{2}}. \end{array}$$□

## 6. Performance Evaluation

In this section, we evaluate the practicality and performance of the scheme PLDP-FL through experiments. We will focus on the impact of privacy parameters on the performance of the scheme and also compare the functionality of the proposed scheme.

### 6.1. Experimental Setup

#### 6.1.1. Datasets

To evaluate the scheme’s practicality more reasonably and comprehensively, we will assess it on synthetic and real datasets. For the synthetic dataset, we randomly generate 100 data records within the safe region selected by the participants, and conduct performance tests on their private data set. Real data sets for experiments are provided by the MNIST and Fashion-MNIST datasets. The MNIST dataset is a handwritten digital image classification data set, containing ten target categories, with 60,000 training samples and 10,000 test samples. The scale, format, and division of the training and test set in Fashion-MNIST are the same as those in MNIST. In contrast to the MNIST dataset, the Fashion-MNIST dataset comprises images depicting ten distinct types of clothing. A convolutional neural network (CNN) is employed for model training, which consists of one input layer, two convolution layers, two max pooling layers and two fully connected layers. The local iteration period is set to 5, the batch size of each iteration is 20, the learning rate is 0.01, and the proportion of participants in each round of training is 0.7.

#### 6.1.2. Metrics

For the synthetic dataset, the relative error (RE) is employed as the evaluation standard of the scheme, and the relative error between the estimated value and the true value is calculated using the following formula.
(15)$$RE=\frac{|T_{v}-E_{v}|}{|T_{v}|},$$
where RE denotes relative error, $T_{v}$ is true value, $E_{v}$ is estimated value. In the context of the MNIST dataset, the most commonly used measure of success is accuracy rate. Our goal is to attain a higher model accuracy while ensuring personalized privacy.

#### 6.1.3. Privacy Parameters

The privacy parameter τ offers personalized privacy protection to local clients, with its safe region representing the range that attackers cannot distinguish from the original data. Thus, the larger the safe region, the more difficult it is to distinguish attackers from the raw data, although it can significantly affect performance. To assess the effect of privacy parameter τ on the scheme, we construct a different safe region size set $L_{\tau }=\left\{0.2,0.4,\ldots ,1.2,\ldots ,2.0\right\}$, each client can choose a size for their safe region, and then adjust their safe region in accordance with the size of the safe region $\tau_{i}$. For example, if a client chooses a safe region size of 1.2, they can set the safe region as [−0.6,0.6],[−0.2,1.0], etc. To accurately evaluate the effect of the safe region on the scheme’s performance, three modes were established: $\tau_{1}$ = {0.2,0.4,…,0.2,0.4}, $\tau_{2}$ = {0.2,0.4,…,1.8,2.0,0.2,…}, $\tau_{3}$ = {1.8,2.0,…,1.8,2.0}, where $\tau_{1}$ being the smallest, $\tau_{2}$ being the next smallest, and $\tau_{3}$ being the largest.

The privacy parameter ϵ also provides personalized privacy protection to the clients, where its size indicates the level of indistinguishability from the adversary. The smaller the ϵ, the more robust the privacy protection, leading to a greater impact on performance. Based on our experience, participants are given the option to choose their privacy budget $\epsilon_{i}$ from the selection set $E_{\epsilon }=\left\{0.1,0.2,\ldots ,0.9,1.0\right\}$. To accurately assess its effect on the scheme’s performance, also set three modes with $\epsilon_{1}$ = {0.1,0.2,…,0.1,0.2}, $\epsilon_{2}$ = {0.1,0.2,…,0.9,1.0,0.1,…}, $\epsilon_{3}$ = {0.9,1.0,…,0.9,1.0}, $\epsilon_{1}$ opts for the least privacy budget for the participants, $\epsilon_{2}$ chooses the privacy budget in ascending order, and $\epsilon_{3}$ selects the highest privacy budget for the participants. We first evaluate the effect of each mode on the model performance, and then assess the effect of their various combinations, such as $\epsilon_{1}\_\tau_{1},\ldots ,\epsilon_{3}\_\tau_{3}$.

#### 6.1.4. Environment

All of our experiments were performed using Python programming language and on a local server (32 Intel(R) Xeon(R) Silver 4110 CPU @ 2.10GHz, 156GB RAM, 2TB, Ubuntu 20.04.3, cuda 11.2, GPU). All test results are obtained by taking the average value after running several times.

### 6.2. Experimental Results

#### 6.2.1. Impact of Privacy Parameters on Synthetic Dataset

For this experiment, 70 participants were used to measure the relative error of the scheme under varying privacy budgets and security intervals. The results of this experiment are illustrated in Figure 2. Figure 2a illustrates the alteration of relative error according to the variation of security interval modes when the privacy budget is changed, while Figure 2b displays the change of relative error in response to the alteration of privacy budget modes when the safe region size is changing. As depicted in Figure 2a, the relative error of the scheme decreases with the increase of the privacy budget. Because a larger privacy budget leads to smaller noise introduced by the local differential privacy algorithm, thus providing more accurate results, which corroborates our prior theoretical analysis. Simultaneously, it is evident from the three security interval mode curves that the larger the security interval, the more significant the effect on the scheme performance, leading to a greater error in the result. As depicted in Figure 2b, the relative error increases with the safe region. However, $\epsilon_{3}$ with a larger privacy budget produces a smaller relative error than $\epsilon_{1}$ with a smaller privacy budget.

#### 6.2.2. Impact of privacy parameters on MNIST and Fashion-MNIST

For the image classification model training experiment of federated learning, we have set 100 local participants and the participation rate for each round of the training process is 0.7. The convolution core is 10*5*5*32, and the first fully connected layer boasts an input channel of 320 and an output channel of 50, while the second fully connected layer has an input channel of 50 and an output channel of 10. It can be observed that the communication cost increases linearly with the increase of the local participant’s proportion from Figure 3a, and Figure 3b illustrates the effect of varying the number of participants on the model’s accuracy. It is evident that the accuracy of the model increases with the number of local participants, due to the fact that a greater amount of intermediate parameters can be shared, thus resulting in a more precise trained model. Figure 3b demonstrates that selecting participants with a ratio of 0.7 or more has a relatively minor impact on the model’s accuracy. To reduce communication costs and maintain accuracy, only 70% of the participants will be selected for training in each round, instead of all participants.

We conducted experiments to observe the effect of privacy parameters on the model’s accuracy in various settings. As illustrated in Figure 4, the accuracy of the model increases with the number of communication rounds. As is evident from Figure 4a,c, when a small privacy budget is chosen, the local differential perturbation algorithm introduces a significant amount of noise to the parameters, resulting in a bad aggregation outcome and a comparatively low accuracy of the final model. In contrast, the model accuracy is notably improved when a larger privacy budget is chosen. These two selection methods are excessively rigorous and do not accurately reflect the personalized privacy requirements of each participant. The $\epsilon_{2}$ mode allows for selecting a privacy budget that can well reflect the individual’s privacy needs, and its final model accuracy is similar to that of the $\epsilon_{3}$ mode. From Figure 4b,d, it is clear that a smaller security interval is set, the accuracy of the resulting model is greater, while a larger safe region yields a lower accuracy. Participants can still obtain a reasonably high model accuracy even with different safe regions. In conclusion, the experiments conducted with different privacy parameters demonstrate that a satisfactory model can be obtained by using personalized privacy parameters with minimal loss.

#### 6.2.3. Impact of Different Privacy Parameters Combinations on MNIST and Fashion-MNIST

We further tested the accuracy of the model when combining different privacy parameters. Each participant can set two privacy parameters according to personalized local differential privacy. The results of three privacy budget selection modes and three safe region selection modes are presented in Figure 5a–f, respectively. Figure 5a,b demonstrate that the combination under $\epsilon_{1}$ mode offers robust privacy protection, though at the cost of low accuracy. Pursuing high-intensity privacy protection, the parameter combination form of $\epsilon_{1}\_\tau_{1}$ can still produce a model with relatively high accuracy after a certain amount of training. As depicted in Figure 5c,e following 50 rounds of training on the MNIST dataset, the accuracy under the combination of modes $\epsilon_{2}$ and $\epsilon_{3}$ can reach 95.7% and 97.9%, respectively. In contrast, it can be seen from Figure 5d,f the highest accuracy achieved by the same combination on the Fashion-MNIST dataset was 79.2% and 85.6%, respectively. Furthermore, the accuracy of the model decreases as the safe region increases, which agrees with the outcomes of the earlier separate examination of privacy parameters. If the data sensitivity of each participant is not high, the $\epsilon_{3}\_\tau_{1}$ parameter combination form can be utilized in order to obtain a model with high accuracy. In case of varying data sensitivities among participants, the parameter combination in $\epsilon_{2}$ mode should be applied, as it offers improved accuracy compared to $\epsilon_{1}$ mode and a reduced accuracy loss compared to $\epsilon_{3}$ mode. Our scheme can produce a model with a high degree of accuracy while still maintaining personalized privacy parameters.

#### 6.2.4. Comparison of Scheme PLDP-FL against the Existing Approaches

The results of our comparison are shown in Figure 6. We compare our scheme with the first, and only federal learning scheme PLU-FedOA [34], which is based on personalized local differential privacy. However, their personalized local differential privacy implementation algorithm differs from ours, as it only has one privacy parameter ϵ. To ensure a fair comparison, the experiment is conducted on MNIST dataseet, and fixed the safe region of τ for each client to [−1, 1] and then compared the performance of its case2 scenario with our $\epsilon_{2}$ mode. The scheme FedAvg [3] can be used as an upper bound for our performance comparison. From Figure 6, we can see that after 100 rounds of training, our scheme has essentially converged, with the accuracy rate of the model reaching 97.5%, which is equivalent to the accuracy rate under non-private conditions. In contrast, the PLU-FedOA scheme only achieved 90.3% accuracy rate. It can be concluded that our scheme is more competitive in terms of model performance.

### 6.3. Functionality Comparison

This section compares the functions implemented by the PLDP-FL scheme with the relevant representative work, and the results are displayed in Table 2. This comparison work focuses on the scheme’s security, such as parameter privacy, model privacy and whether it can prevent inference attacks. It also looks into the practicability of the scheme, such as the size of the actual privacy budget parameter when differential privacy is applied. Furthermore, it assesses whether the scheme supports personalized privacy protection of local clients. The reference [3] introduced the federated learning method of model aggregation, yet this method fails to provide privacy protection for the data during the training process, thus making it insecure. The reference [19] produced a new algorithm to enhance privacy loss by utilizing global differential privacy technology with a random gradient descent algorithm. Unfortunately, this technique is not compatible with distributed learning scenarios. The scheme [12] introduced a new federated learning system using CLDP, a variant technology of local differential privacy, which addresses the problem that the existing local differential privacy protocol is unsuitable for high-dimensional data. However, the actual corresponding privacy budget in the scheme is large, resulting in a weak privacy protection effect, making it unfeasible. Additionally, the local participants in the scheme cannot gain personalized privacy protection. A noise-free differential privacy mechanism for the federated model distillation framework was proposed in [26], which can solve the training data leakage caused by model prediction. However, it fails to provide personalized privacy protection to the local participants. Our proposed approach commences by addressing participants’ individualized privacy protection requirements, constructing a personalized local differential privacy algorithm that satisfies these requirements, and training superior models while providing personalized privacy protection for participants.

## 7. Conclusions

In this paper, we present a personalized local differential privacy federated learning scheme, to overcome the limitations of the existing local differential privacy-based federated learning scheme. This scheme has developed a perturbation algorithm PDPM, to satisfy the personalized local differential privacy needs of local participants. The PDMP algorithm enables each participant to decide their own privacy parameter pair $\left(\tau_{i},\epsilon_{i}\right)$, instead of the fixed privacy parameter ϵ, thus satisfying the personalized privacy protection requirements in the federated learning process. Numerous experiments have demonstrated that clients can adjust their privacy parameters while still obtaining a high-accuracy model. This paper only focuses on the personalized privacy protection needs of the client. In fact, every participant’s data may contain multiple attributes, and each attribute’s privacy sensitivity might differ. and the value range of each attribute may contain multiple values, and the privacy sensitivity of each value may also differ. In the future, we intend to design solutions that address personalized privacy protection needs from multiple angles, such as the attributes and values of the client.

## Figures and Tables

**Figure 1 entropy-25-00485-f001:**
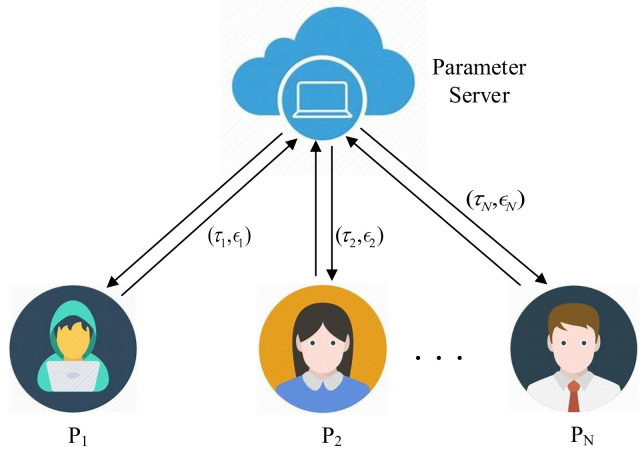
PLDP-FL Framework.

**Figure 2 entropy-25-00485-f002:**
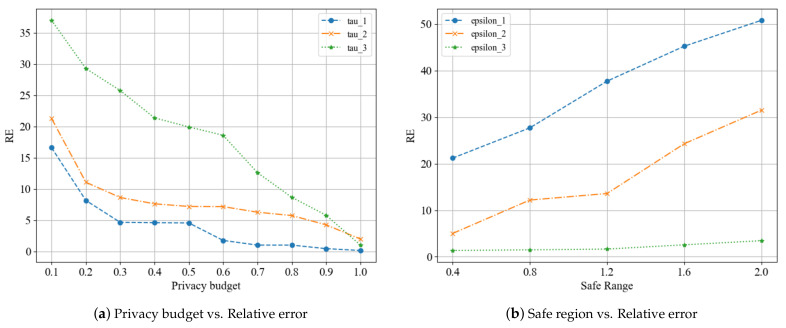
Relative error of the scheme under different privacy parameters.

**Figure 3 entropy-25-00485-f003:**
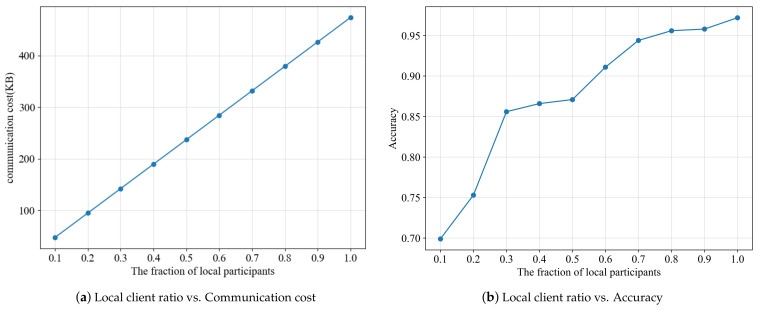
Performance of the scheme under different local client ratio.

**Figure 4 entropy-25-00485-f004:**
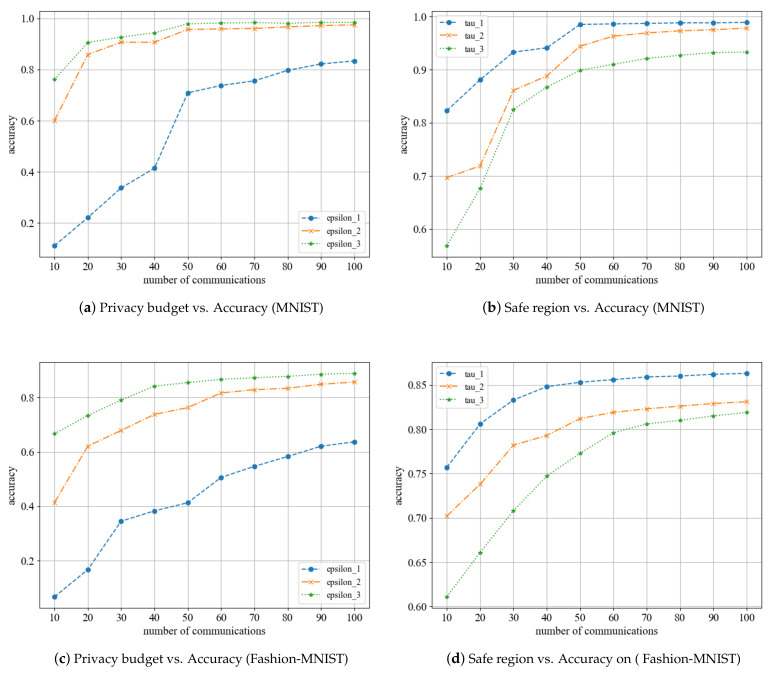
Performance of the scheme under different parameters.

**Figure 5 entropy-25-00485-f005:**
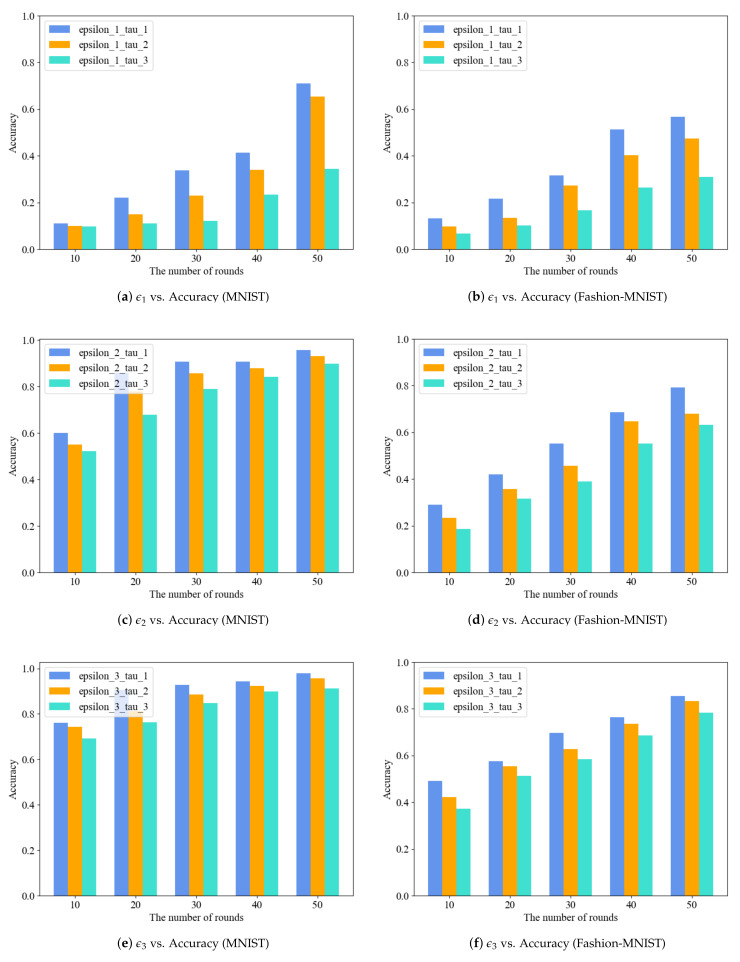
Classification accuracy of scheme PLDP-FL on image dataset.

**Figure 6 entropy-25-00485-f006:**
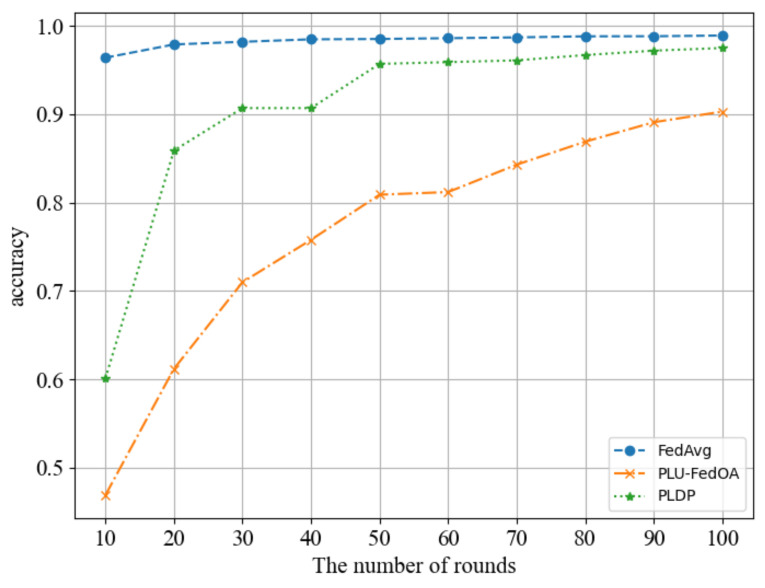
Comparison of scheme PLDP-FL against the existing approaches.

**Table 1 entropy-25-00485-t001:** Summary of Main Notation.

M	A randomized algorithm for DP
$v,v^{\prime }$	Any pair of input values
$\tilde{v}$	The perturbed value of the input value *v*
$\epsilon_{i},\tau_{i}$	The privacy parameters of client $P_{i}$
$L_{i},c_{i}$	The length and midpoint of the safe range, respectively
$P_{i}$	The *i*-th client
$D_{i}$	The dataset held by client $P_{i}$
$x_{i},y_{i}$	The input data and true label in the dataset $D_{i}$, respectively
*N*	The number of all clients
$|P_{t}|$	The number of chosen clients
γ	The local client selection factor
*T*	The number of aggregation rounds
L	Loss function
$w^{*}$	The optimal model parameters that minimize L
$w_{i,t}$	Local parameters of the *i*-th client in the *t*-th round
${\tilde{w}}_{i,t}$	Local parameters after perturbation of the *i*-th client in the *t*-th round
${\bar{w}}_{t}$	Global parameters after aggregation

**Table 2 entropy-25-00485-t002:** Functionality Comparison.

Scheme	Techniques	Parameter Privacy	Model Privacy	Practical	Protection against Inference Attacks	Personalized Privacy Protection
FedAvg [3]	FL	×	×	-	×	-
DPSGD [19]	DP	✓	✓	✓	✓	-
LDP-Fed [12]	LDP	✓	✓	×	✓	×
FedMD-NFDP [26]	NFDP	✓	✓	✓	✓	×
PLU-FedOA [34]	PLDP	✓	✓	×	✓	✓
PLDP-FL	PLDP	✓	✓	✓	✓	✓

## Data Availability

Not applicable.

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
