# Peer review of "PLDP-FL: Federated Learning with Personalized Local Differential Privacy"

_entropy, 2023, doi:10.3390/e25030485_

Round 1

Reviewer 1 Report

This article proposes the PLDP-FL: Federated Learning with Personalized Local Differential Privacy scheme to overcome the limitations of the existing local differential privacy-based federated learning scheme. They introduced two algorithms for the proof of concept and demonstrated their proposal through a mathematical model and simulations.

The authors should address the recommendations I have noted in their submitted article, mainly about their contributions section (Please refer to my comments on the document.)

Author Response

Thank you for your suggestion. Please refer to the attachment for comments on revision.

Reviewer 2 Report

Shen et al proposed a personalized local differential privacy data perturbation mechanism for a federated learning scheme. Overall, the research question is important and interesting. However, it is unclear how the proposed method can decrease the computation and communication costs without affecting performance. The authors should also propose potential application scenarios of the proposed method, e.g., cross-silo or cross-device. 

1. The authors randomly selected a certain proportion for parameter sharing to decrease the communication cost. However, Fig. 4 (d) shows that this scheme hurts the performance of the model. The authors should add a figure to illustrate the extent of the decreased communication cost. 

2. The authors should demonstrate how to determine the optimal privacy parameter to ensure privacy and maintain the model’s accuracy, i.e., how to balance the trade-off between communication costs and model performance.

2. One concern for Fig. 3 (a), Fig. 3 (b) and Fig. 5: it seems that the model is not converged well. The authors should train more epochs and show the training curve.

3. The authors state that the differential privacy methods need less communication cost and computation power compared to Cryptography-based schemes. What about the accuracy of these two paradigms? Also, the authors can consider comparing with the Cryptography-based method in Fig. 5.

4. The merit of this study is providing personalized privacy protection to the clients, which allows them to define their privacy parameters to perturb the updated weights before uploading. However, the authors should compare the differences between the proposed study and existing ones (e.g., Yang et al, Federated Learning with Personalized Local Differential Privacy, 2021) and discuss the limitations of the proposed method.

6. The authors should discuss the potential application fields and how to apply the proposed FL when there is device heterogeneity in the real world.

Author Response

(The authors gave the same response as above.)

Reviewer 3 Report

The paper explores the topic of Federated Learning with Personalized Local Differential Privacy. The authors present a new perturbation algorithm called PDPM, which satisfies the concept of PLDP. 

This allows clients to tailor their privacy parameters to the sensitivity of their data, providing a more personalized and secure form of privacy protection.

The results of experiments conducted on this scheme demonstrate its efficacy in producing high-quality models while also meeting the unique privacy requirements of individual clients.

The Limitations of the Scheme:

• The scheme is limited to addressing only the personalized privacy protection needs of clients and does not consider other factors such as attributes or values.

• While the paper has been tested using both synthetic and real-world data sets, further testing is necessary to ensure its practical application in a wider range of scenarios.

I believe that the references should be added to the main body of the paragraph rather than remaining in the section titles (see, for example, 2.3.1–2.3.2).

 Accept with minor revisions.

Author Response

(The authors gave the same response as above.)

Round 2

Reviewer 2 Report

No further comments